# Do hypnotics increase the risk of driving accidents or near miss accidents due to hypovigilance? The effects of sex, chronic sleepiness, sleep habits and sleep pathology

Sylvie Royant-Parola[1], Viviane Kovess[2], Agnès Brion[1], Sylvain Dagneaux[1], Sarah Hartley[1,3]*

1 Réseau Morphée, Garches, France, 2 EA 4057, Paris Descartes University, Paris, France, 3 EA 4047, APHP Hôpital Raymond Poincaré, Sleep Center, Université de Versailles Saint-Quentin en Yvelines, Garches, France

* sarah.hartley@reseau-morphee.fr

**Data Availability Statement:** All relevant data are within the manuscript and its Supporting Information files.

## Abstract

Driving accidents due to hypovigilance are common but the role of hypnotics is unclear in patients suffering from sleep disorders. Our study examined factors influencing accidents and near miss accidents attributed to sleepiness at the wheel (ANMAS). Using data from an online questionnaire aimed at patients with sleep disorders, we analysed the associations between ANMAS, sociodemographic data, symptoms of sleep disorders, severity of insomnia (Insomnia Severity Index (ISI)) symptoms of anxiety and depression (Hospital Anxiety and Depression scale with depression (HADD) and anxiety (HADA) subscales), chronic sleepiness (Epworth sleepiness scale ESS), hypnotic use and information about sleep habits. Hypnotics were hierarchically grouped into Z-drugs, sedative medication, melatonin and over the counter (OTC) alternative treatments. Of 10802 participants; 9.1% reported ANMAS (Men 11.1% women 8.3%) and 24.4% took hypnotics (Z-drugs 8.5%, sedative medication 8%, melatonin 5.6% and alternative treatments 2.5%). Logistic regression analysis identified the following risk factors for ANMAS: moderate (OR 2.4; CI: 2.10–2.79) and severe sleepiness (ESS OR 5.66; CI: 4.74–6.77), depression (HADD OR 1.2; CI: 1.03–1.47), anxiety (HADA OR 1.2;CI: 1.01–1.47), and insufficient sleep (OR1.4; CI: 1.2–1.7). Hypnotics were not associated with an increased risk of ANMAS in patients suffering from insomnia. Risk factors varied according to sex: in females, sex (OR 0.; CI: 0.55–0.74), mild insomnia (OR 0.5; CI: 0.3–0.8) and use of alternative treatments (OR 0.455, CI:0.23–0.89) were protective factors and risk was increased by sleepiness, sleep debt, social jetlag, caffeine use, anxiety and depression. In men no protective factors were identified: sleepiness, sleep debt, and severe insomnia were associated with an increased risk of ANMAS. In clinical practice, all patients with daytime sleepiness and men with severe insomnia should be counselled concerning driving risk and encouraged to avoid sleep debt.

**Funding:** This work was supported by the French sleep research society (SFRMS: Société Française pour la Recherche et Médecine du Sommeil) which contributed to the original development of the online questionnaire. The Réseau Morphée is funded by the Paris region health authority (ARS Ile de France). No specific grant for the study was obtained from any funding agency in the public, commercial or not-for-profit sectors.

**Competing interests:** The authors declare no conflicts of interest.

# Introduction

Sleepiness at the wheel is an important cause of traffic accidents, with estimates of the proportion of accidents due to hypovigilance ranging from 3–37%. The relative proportion of accidents due to hypovigilance in France is increasing as road safety campaigns sucessfully reduce speeding and driving under the influence of alcohol and drugs [1–3]. Many factors associated with accidents due to hypovigilance at the wheel have been identified by epidemiological studies. Contributing factors are sleep deprivation, poor quality, fragmented or non-restorative sleep due to external factors, medical or sleep disorders (such as sleep apnea) and the use of sedatives such as alcohol or medication [3–7].

## Sleepiness

Sleepiness can be objectively measured electrophysiologically or defined subjectively, and scales have been developed to quantify subjective sleepiness either at a point in time or over a period. While experimental studies have looked at objective measures of sleepiness while driving, the majority of studies on sleepiness at the wheel have used self reported sleepiness [8]. However the association between sleepiness and accident risk is not straightforward and depends on the scale used and the driver population: professional drivers only show increased risk when subjective sleepiness is severe [2,9–11]. Sleepiness can be due to inadequate or fragmented sleep and is influenced by circadian phase, explaining why accident risk is higher at night [8]. Both acute and chronic sleep deprivation have been shown to be associated with accident risk [6,12]. Sleep-wake time variability, often termed social jetlag, is expressed as the variability in midsleep time and has been suggested to weaken the circadian signal, but its effect on chronic sleepiness measured by the Epworth Sleepiness Scale(ESS) remains unclear [13]

## Sleep disorders

Patients with sleep disorders have an increased accident risk which has been highlighted in several studies [3,9,11,12,14,15]. In the case of obstructive sleep apnea, sleepiness is considered to play an important role, but this increased risk is also seen in insomnia. Patients with insomnia often display hypervigilance with difficulties falling asleep at night and also report that they never nap during the day. This suggests that their increased accident risk may not be solely due to hypovigilance [16]. Driving safely is a complex skill, and it has been suggested that subtle deficits in cognitive and psychomotor performance may underpin the increased accident risk in insomniacs. Several studies have failed to demonstrate modifications in cognitive and psychomotor performance in insomniacs [16–18], but tests are often individually brief. Driving requires sustained attention over a prolonged period of time and a frequent complaint of insomniacs is difficulty in concentrating on a task after a poor nights sleep. Studies of sustained attention in insomniacs over 20 minutes [19] do show deficits which may contribute to the increased accident risk. Both anxiety and depression are prevalent in patients with sleep disorders but have also been shown to independently increase accident risk [20,21].

## Hypnotic use

Hypnotics increase sleepiness. Several epidemiological studies have highlighted an association between the use of benzodiazepines or "Z-drugs" (Zopiclone and Zolpidem) and increased accidents [22–25]. A review of 27 controlled studies confirmed an increased accident risk for Zolpidem, Zopiclone, Temazepam, Diazepam, Flunitrazepam, Flurazepam, Lorazepam, and Triazolam [26] at least during the first 2–4 weeks of treatment. Studies using healthy

volunteers and driving simulators have confirmed this risk, which has been shown to be dependent on the dose, time between the dose and the test and the half-life of the hypnotic [27–29]. Accident risk in experimental studies is generally performed with driving simulators, which have been shown to be consistent with on-road driving test results [30].

Patients with insomnia take hypnotics in order to improve night time sleep: either by reducing sleep latency and/or improving sleep continuity. If the increased accident risk in insomniacs is due to reduced sleep time or poor quality sleep, it has been suggested that improved sleep following treatment by hypnotics could therefore reduce risk although this would be limited by the known effects of hypnotics on daytime vigilance. The results of studies are conflicting: performance on a driving simulator in insomniacs has been shown to be negatively affected by the use of long half-life but not short half-life hypnotics even after 7 days of use [28], and on-road driving studies in older patients with insomnia show that chronic habitual hypnotic use does not affect driving performance [31] although the introduction of a new hypnotic increases risk [32] probably as tolerance to daytime effects has not taken place [25].

### Gender

Driving accidents are more common in men [15] even when controlling for mileage and the higher proportion of male drivers [33]. A study of accidents in patients with OSA confirms that even when controlling for severity of sleep apnea, both near miss accidents and accidents are less frequent in women, even in women with severe sleepiness, suggesting that women may react differently to the sensation of sleepiness [11]. However, the factors underlying this increased risk have been little studied.

### Disadvantages of studies to date

Determining the contribution of hypnotics to the risk of accidents has been complicated by the fact that most of the experimental studies have been performed with healthy volunteers. In healthy volunteers, hypnotics have no potential to improve nighttime sleep and thus daytime functioning [26,32] and the results will reflect the residual daytime sedating effects of the treatment. In addition, most studies are performed with the occasional use of hypnotics, often after a period of washout in patients previously using hypnotics, which does not reflect the chronic use observed in most populations [34]. Rapid habituation to the daytime sedating effects of hypnotics is observed both experimentally [32] and in clinical practice, while subjective benefits on sleep quantity and quality are conserved even if studies do not show continued objective improvements in sleep [35].

### Hypothesis and aim

Our hypothesis was that factors known to be associated with increased daytime sleepiness such as sleep apnea, sleep debt and the use of hypnotics would be associated with ANMAS.

Our objective was to explore factors associated with accidents and near miss accidents attributed to sleepiness at the wheel (ANMAS).

## Materials and methods

### Sample

(Table 1). All responders aged over 18 to an online questionnaire from 19/06/2018 to 26/04/2019 were included. Complete data was available for 10802 participants of whom 28% were men. The mean age of participants an age of 41.9 ±14.1 years (range 18–93.6).

Table 1. Participants: Socio-demographic data, symptoms, sleep pathology, sleep habits and hypnotic use.

| | | ANMAS in the last 6 months | | | |
| --- | --- | --- | --- | --- | --- |
| | | Yes | No | p | Total |
| Population% (Total number) | | 9% (984) | 91% (9818) | | 100% (10802) |
| Age mean ±SD | | 40.92 ±13.3 | 42.06 ±14.1 | 0.008 | 41.9 ± 14.1 |
| Male% | | 11.1 | 88.9 | <0.0001 | 3064 (100%) |
| Female % | | 8.3 | 91.7 | | 7738 (100%) |
| BMI mean ±SD | | 25.43 ±5.2 | 24.8 ±5.3 | 0.0002 | 24.8 ±5.2 |
| Risk of sleep apnea % | | 11% | 7% | <0.0001 | 7% |
| ESS mean ±SD | | 12.7 ±5 | 9 ±4.7 | <0.0001 | 9.4 ± 4.9 |
| ISI mean ±SD | | 17.5 ±5.5 | 16.4 ±5 | 0.01 | 16.5 ±5.1 |
| HADA mean ±SD | | 9.7 ±4 | 9.5 ± 3.9 | 0.007 | 9.6 ±4 |
| HADD mean ±SD | | 7 ±4 | 6.4 ±3.9 | <0.0001 | 6.5 ±3.9 |
| Habits | Coffee mean number of cups ±SD | 3.5 ±1.9 | 3.2 ±1.8 | 0.0001 | 3.2 ±1.8 |
| | Alcohol mean number of glasses ±SD | 2 ±1.7 | 1.8 ±1.3 | 0.0224 | 1.9 ±1.3 |
| | Time in bed in the week mean ±SD | 7.72 ±1.4 | 8 ±1.5 | <0.0001 | 8 ±1.4 |
| | Sleep debt during the week% | 24% | 15% | <0.0001 | 16% |
| | Social jetlag % | 48% | 41% | <0.0001 | 42% |
| Use of hypnotic for sleep* | 0—None % | 78.6% | 75% | 0.010 | 76% |
| | 1—Z-drugs % | 8.2% | 8.5% | | 8.5% |
| | 2—Sedating medication % | 6.5% | 8% | | 8% |
| | 3—Melatonin % | 5.6% | 5.6% | | 5.6% |
| | 4—Alternative treatments % | 1% (10) | 2.6% (255) | | 2.5% |

Results are given in % (total number) or mean ±SD

*Treatment groups are hierarchized (see Methods)

## Procedure

Participants completed an online questionnaire, aimed at French speaking patients suffering from sleep disorders and accessible via the Réseau Morphée website which offers high quality, non-commercial information about sleep disorders. The website was set up in 2004 and is well known in France. The questionnaire was added to the site in 2017 and is a key tool in the patient care pathway, allowing patients to evaluate sleep symptoms before a medical consultation. Apart from an initial press conference, no publicity was needed to recruit participants. Specific questions concerning hypnotic use were added to the questionnaire on the 19/06/2018, at the request of the regional health authority. Access to the questionnaire is free and participants gave their consent to the use of data for research. The study was approved by the scientific committee of the Réseau Morphée and by two patient associations, France Insomnie and Sommeil et Santé. As a non interventional study (MR004) it was approved by the CNIL (Commission Nationale Informatique et Liberté, 8013081 19/12/2016). Only data from participants over the age of 18 was included in the study.

## Methods

Accident risk was explored by asking whether the patient had had a driving Accident or a Near Miss Accident due to Sleepiness (ANMAS) over the past six months. Questions examined sociodemographic data, information about symptoms of sleep disorders and this was quantified where possible with validated scales. Chronic sleepiness was measured using the Epworth Sleepiness Scale ESS [36] which evaluates sleepiness over the past two weeks and is extensively

used by sleep physicians. A score >10 is indicative of excessive daytime sleepiness. Insomnia was measured using the Insomnia Severity Index ISI [37] which evaluates both nighttime and daytime symptoms of insomnia. A score >7 identifies insomnia and >14 moderate to severe insomnia. A potential risk of sleep apnea (OSA) was defined as the presence of snoring and respiratory pauses.

Sleep habits were assessed by asking participants about bed and wake times. Potential sleep deprivation was defined as a deficit of >1 hour between declared sleep needs and mean estimated time in bed on weekdays. Social jet lag was calculated as a ≥3 hours difference in sleep midpoint between the week and weekends [38].

Mental health symptoms were measured by Hospital anxiety and Depression scale HAD which contains two subscales measuring anxiety and depression. These dimensions were analysed separately as anxiety may be associated with hypervigilance and depression with daytime sleepiness [39]. Data on present hypnotic use identified patients using sleeping medication and specified the use of Zopiclone, Zolpidem, melatonin (in prescribed and in over the counter formulations), hydroxyzine (available over the counter in France), herbal medications and homeopathy and other prescribed sleeping medications; multiple answers were possible. To take multiple therapies into account, sleep treatments were grouped into four mutually exclusive hierarchical categories: category 1) Z-drugs (Zopiclone and Zolpidem), category 2) sedative treatments (hydroxyzine and prescribed hypnotics other than Z-drugs) and no category 1, category 3) melatonin in both OTC and prescribed formulations and no categories 1or 2, and finally category 4) Herbal medications/homeopathy/dietary supplements and no other categories.

## Analysis

Data was collated and cleaned in Excel, and analysis was performed using STATA 13.1. The HAD was analysed according to the two subscales: depression and anxiety: scores >10 were considered to be indicative of significative anxiety or depressive symptoms. Daytime sleepiness was analysed according to the Epworth sleepiness scale as a continuous variable and further categorized into no sleepiness (ESS 1–10), moderate sleepiness (ESS 11–15) and severe sleepiness (ESS >15). The score on the ISI was grouped into no insomnia (0–7), mild insomnia (ISI 8–14), moderate insomnia (ISI 15–21) and severe insomnia (ISI 22–28). Analysis was performed using Chi squared tests for discrete variables and variance analysis to compare means. Multiple and logistic regression was used for continuous or categorical variables in order to establish odds ratios.

## Results

### Population description

Participants were included from 19/06/2018 to 26/04/2019. Complete data was available for 10802 participants of whom 28% were men with a mean age of 41.9 ±14.1 years. 9.11% reported ANMAS (Table 1).

### Sleep complaints and psychiatric symptoms

The majority of participants suffered from insomnia with a mean ISI score of 16.5 ± 5.1: only 4.77% had an ISI <7. 24.3% had mild insomnia, 56.5% had moderate insomnia and 14.4% severe insomnia. The mean HADD score was 6.5 ± 3.9 and 16.9% had an HADD score >10 compatible with clinical depression. The mean HADA score was 9.6 ± 4 and 41.5% had an HADA score >10 compatible with clinical anxiety.

## Sleep habits and behaviours

Participants spent a mean of 8 hours in bed during the week, but 16% reported a sleep debt of at least one hour on week nights compared to their estimated sleep need. This was reflected in the prevalence of significant social jetlag (with a midpoint difference of 2 hours or more) in 42%. Participants consumed a mean of 3.2 cups of coffee or other stimulating beverages per day and 1.9 glasses of wine.

## Use of hypnotics

Overall 76% of patients took no hypnotic treatment. Hypnotic treatments included Zolpidem (17.9%), Zopiclone (20.2%), over the counter (OTC) hydroxyzine (21.8%), prescribed melatonin (21.8%), OTC melatonin (35.9%), OTC herbal medications and homeopathy (19.9%) and various OTC dietary supplements (31.1%). Treatments could be combined: 56.1% were on monotherapy, 25.9% two treatments, 12.3% three and 5.7% four or more. In order to account for the use of multiple treatments, Table 1 reports the use of hypnotic treatments by mutually exclusive hierarchical categories: overall, 8,5% took only Z-drugs (Zopiclone and Zolpidem), 8% took sedative treatments (hydroxyzine and prescribed hypnotics other than Z-drugs), 5,6% took melatonin in both OTC and prescribed formulations, and 2,5% took only alternative therapies (Herbal medications/homeopathy/dietary supplements).

## Univariate analysis

ANMAS were significantly more common in men (11.1% vs 8.3%, p<0.0001) and in younger participants (mean age 40.9 versus 42.1 p = 0.0083).

## Sleep complaints and psychiatric symptoms

ANMAS were significantly associated with increased daytime sleepiness with a mean score on the ESS (12.2 ± 4.9 vs 8.6 ± 4.7 p<0.0001). Insomnia symptoms were also more frequent: small but significant differences were shown in mean ISI scores (16.5 ± 5.5 vs 16.3 ± 5.1, p = 0.01). Patients at risk of sleep apnea had more ANMAS (11% vs 7%; p<0.0001). Psychiatric symptoms were more common in patients reporting ANMAS with an increase in both anxiety symptoms (HADA score 9.7± 4 vs 9.5±3.9; p = 0.007) and depressive symptoms (HADD score 7±4 vs 6.4 ±3.9; p<0.0001).

## Sleep habits and behaviour

Sleep debt during the week was significantly more common in participants with an increased accident risk (20% vs 15%, p<0.0001) as was social jetlag (44% vs 40%, p<0.0001). Alcohol consumption was increased In ANMAS (2 glasses/day ±1.7 vs 1,8 glasses/day ±1.3; p = 0.02) as was the use of caffeine containing drinks (3,5 cups/day ±1,9 vs 3,2 cups/day ±1,8; p = 0.0001).

## Hypnotic use

The risk of ANMAS was reduced in participants taking any medication for sleep: 7.9% versus 9.5% (p = 0,021). Table 1 shows that hypnotic consumption by hierarchical category was lower in patients with ANMAS: Z-drugs 8.2% vs 8.5%, sedating medication (6.5% vs 8%) and alternative treatments (1% vs 2.6%) and identical in the melatonin group (5.6% vs 5.6%).

## Factors associated with ANMAS

(Multivariate regression analysis: Table 2).

**Table 2. Logistic regression for ANMAS (n = 10,494).**

| | Odds Ratio | P>\|z\| | 95% Conf. Interval | |
|---|---|---|---|---|
| Sex (female/male) | 0.639 | 0.000 | 0.550 | 0.744 |
| Age (continuous) | 0.998 | 0.453 | 0.992 | 1.003 |
| Possible sleep apnea/no sleep apnea | 1.144 | 0.261 | 0.905 | 1.445 |
| Mild insomnia (ISI 8–14)/<8 | 0.752 | 0.109 | 0.530 | 1.066 |
| Moderate insomnia (ISI 15–21) /<8 | 1.01 | 0.945 | 0.730 | 1.401 |
| Severe insomnia (ISI >21) /<8 | 1.323 | 0.129 | 0.921 | 1.901 |
| Anxiety (HADA >10)/ = <10 | 1.174 | 0.035 | 1.011 | 1.362 |
| Depression (HADD >10)/ = <10 | 1.231 | 0.020 | 1.033 | 1.468 |
| Moderate sleepiness (ESS 11–15)/<11 | 2.369 | 0.000 | 2.012 | 2.789 |
| Severe sleepiness (ESS>15)/<11 | 5.663 | 0.000 | 4.739 | 6.768 |
| Sleep debt during the week/no debt | 1.432 | 0.000 | 1.209 | 1.696 |
| Social jetlag/no jetlag | 1.148 | 0.072 | 0.988 | 1.333 |
| Caffeine (> 3 cups / day)/ = <3 | 1.207 | 0.042 | 1.006 | 1.448 |
| 1.Z drugs | 0.971 | 0.890 | 0.890 | 1.469 |
| 2 Sedating medication | 0.765 | 0.151 | 0.531 | 1.102 |
| 3 Melatonin | 0.986 | 0.942 | 0.664 | 1.462 |
| 4 Alternative treatments | 0.455 | 0.022 | 0.231 | 0.893 |

*Treatment groups are hierarchized (see Methods), Drug treatments by group are compared to no treatment

A model controlling for age and sex, sleep and psychiatric symptoms (potential OSA, insomnia, depression and anxiety), symptoms of hypovigilance (ESS), sleep habits (sleep debt, social jet-lag, excessive caffeine consumption (>3 cups) and the use of different treatments (type and number) showed a significantly increased risk of ANMAS in patients with daytime sleepiness, especially severe sleepiness with an ESS >15 (OR 5.66, CI 4.74–6.77; p<0.0001), patients with depression (HADD>10 OR 1.23 CI 1.03–1.47; p = 0.02) and anxiety (HADA>10 OR 1.17 CI 1.011–1.36; p = 0.035), sleep debt (OR 1.432, CI 1.209–1.696; p<0.0001), and high caffeine consumption (OR 1.207, CI 1.00–1.45; p = 0.042). Protective factors include female sex (OR 0.64, CI 0.55–0.74), and specifically use of OTC herbal medications/homeopathy/dietary supplements as the only treatment (OR = 0.45, CI 0.23–0.89). No effect was found for age, the consumption of Z-drugs, sedating medication or melatonin.

### Factors associated with ANMAS by sex

(Multivariate regression analysis Table 3).

ANMAS associated factors differed between men and women. Chronic sleepiness remained relevant in both groups although the odds ratio for severe sleepiness with an ESS>15 was larger in men (OR 6.98 CI 5.12–9.53; p<0.0001) than in women (OR 5.14, CI 4.13–6.40; p<0.0001). Mild insomnia was protective in women (OR 0.53, CI 0.34–0.84; p = 0.006) but not in men while severe insomnia was a risk factor in men (2,151 CI 1,165–3,970) but not in women. Social jet-lag (OR 1.201, CI 1.00–1.44; p = 0.05) and caffeine consumption (OR 1.282, CI 1.01–1.62; p = 0.038) were risk factors only in women. No hypnotic significantly increased risk, but alternative treatments reduced risk only in women (OR 0.414, CI 0.2–0.88; p = 0.02).

### Discussion

This large questionnaire based study of patients reporting sleep disorders found that reported ANMAS over the preceding 6 months were increased in men, by the presence of daytime

**Table 3. Logistic regression for ANMAS (n = 10,494) in women compared to men.**

| | Women n = 7.521 | | | | Men = 2.973 | | | |
|---|---|---|---|---|---|---|---|---|
| | Odds Ratio | P>\|z\| | 95% Conf. Interval | | Odds Ratio | P>\|z\| | 95% Conf. Interval | |
| Age (continuous) | 0.996 | 0.273 | 0.989 | 1.003 | 1.000 | 0.982 | 0.990 | 1.009 |
| Presence of relevant pathology | | | | | | | | |
| sleep apnea/no sleep apnea | 1.014 | 0.938 | 0.709 | 1.452 | 1.189 | 0.286 | 0.865 | 1.633 |
| Mild insomnia (ISI 8–14)/<8 | 0.534 | 0.006 | 0.340 | 0.838 | 1.205 | 0.514 | 0.689 | 2.106 |
| Moderate insomnia (ISI 15–21) /<8 | 0.7878 | 0.257 | 0.522 | 1.190 | 1.491 | 0.142 | 0.874 | 2.544 |
| Severe insomnia (ISI >21) /<8 | 0.992 | 0.973 | 0.631 | 1.559 | 2.151 | 0.014 | 1.165 | 3.970 |
| Anxiety (HADA >10)/ = <10 | 1.21 | 0.035 | 1.014 | 1.451 | 1.116 | 0.427 | 0.851 | 1.465 |
| Depression (HADD >10)/ = <10 | 1.306 | 0.012 | 1.059 | 1.610 | 1.084 | 0.629 | 0.783 | 1.500 |
| Symptoms of hypovigilance | | | | | | | | |
| Moderate sleepiness (ESS 11–15)/<11 | 2.318 | <0.0001 | 1.897 | 2.833 | 2.385 | <0.0001 | 1.797 | 3.166 |
| Severe sleepiness (ESS>15)/<11 | 5.138 | <0.0001 | 4.126 | 6.398 | 6.981 | <0.0001 | 5.117 | 9.525 |
| Sleep habits | | | | | | | | |
| Sleep debt during the week/no debt | 1.387 | 0.002 | 1.130 | 1.702 | 1.522 | 0.006 | 1.125 | 2.059 |
| Social jetlag/no jetlag | 1.201 | 0.049 | 1.001 | 1.441 | 1.054 | 0.699 | 0.807 | 1.376 |
| Caffeine (> 3 cups / day)/ = <3 | 1.282 | 0.038 | 1.014 | 1.621 | 1.118 | 0.45 | 0.837 | 1.495 |
| Use of sleep treatment* | | | | | | | | |
| 1.Z drugs | 0.817 | 0.427 | 0.496 | 1.345 | 1.993 | *0.09* | 0.897 | 4.429 |
| 2 Sedating medication | 0.710 | 0.111 | 0.466 | 1.082 | 1.173 | 0.687 | 0.540 | 2.549 |
| 3 Melatonin | 0.911 | 0.687 | 0.579 | 1.433 | 1.625 | 0.253 | 0.707 | 3.738 |
| 4 Alternative treatments | 0.414 | 0.022 | 0.195 | 0.882 | 0.707 | 0.657 | 0.154 | 3.260 |

*Treatment groups are hierarchized (see Methods), Drug treatments by group are compared to no treatment

sleepiness, depression, sleep debt during the week and excess caffeine consumption. We did not find that hypnotic treatment increased the risk of ANMAS, on the contrary, we found a protective effect of alternative hypnotic treatments. This study expands on the existing literature by examining the role of hypnotics in AMNA in a large sample size, using a measure of insomnia severity and is the first study to compare accident risk profile between men and women.

Our study confirms that ANMAS are frequent. 9% of our participants reported an ANMAS over the past 6 months which is comparable with other studies: an international study found that 9% of insomniacs had fallen asleep at the wheel and 4% had an accident over the past 12 months [40], and a French study found 5.8% of the general population over the past 12 months reported an accident due to sleepiness [9].

## ANMAS and sleep complaints

We confirmed that ANMAS were strongly linked to chronic sleepiness as measured by the ESS. Our univariate analysis showed an association between ANMAS and symptoms of insomnia, possible sleep apnea, anxiety and depression, confirming the results of previous studies which also found an increased accident risk in patients suffering from sleepiness, probable sleep apnea [2,3,7,9], depression and anxiety [14]. However we note that controlling for chronic sleepiness in our sample removed the increased risk associated with sleep apnea and insomnia, implying that the increased risk of ANMAS in patients with sleep pathology is related to increased chronic daytime sleepiness.

## ANMAS and sleep habits

Insufficient sleep has been identified as a risk factor for driving accidents [6,7,9] and the importance of a sleep debt was confirmed by our model, even after controlling for sleepiness. Sleep debt in our study was defined as a mean time in bed on weekdays which was at least an hour less than estimated sleep needs. Sleep debt is often behavioral and due to insufficient time in bed but may also be due to a shortened sleep time caused by insomnia. However not all insomniacs have insufficient sleep. Insomniacs frequently underestimate sleep time, and in addition may lengthen their time in bed as a compensatory measure. By asking about time in bed, we focused on available time for sleep whether the participant was asleep or not. If time in bed is considerably less than perceived sleep need, this represents a measure of partial sleep deprivation. Chronic partial sleep deprivation has been shown to lead to modifications in executive function [41], to increase impulsive actions [42] and to reduce sustained [43] and vigilant attention [44] which could increase ANMAS even in the absence of sleepiness. Insomniacs can be divided into two groups, those with and those without reduced sleep time and our finding that sleep debt contributes to ANMAS implies an increased risk in those with reduced sleep time. Caffeine is often used by patients to increase vigilance, although excessive consumption can fragment sleep and reduce sleep quality [45]. We note that the consumption of excess caffeine (>3 cups a day) is higher in patients with ANMAS and that this increased risk persisted in our controlled model but only in women.

**ANMAS and hypnotic use.**   We expected to find an association between hypnotic use, especially Z drugs and other sedative medication, and ANMAS and were surprised to find that this was not the case. Hypnotics are prescribed to treat sleep onset and sleep maintenance insomnia. While benzodiazepines may also be prescribed for anxiety, this is not the case for Z-drugs or for melatonin. Insomniacs complain of a reduction in sleep quantity and quality, and hypnotic use has been shown to improve subjective and objective sleep quantity and quality, at least in the short term (<1 month) although after a year there is no difference in objective and subjective sleep quality in patients treated by zopiclone [35]. We found no increase in ANMAS in patients taking Z-drugs or sedating hypnotics which is reassuring given their frequent use for insomnia, and we suggest that this is linked to the improved quality and quantity of sleep which outweigh daytime sedative effects. While hypnotics with a long half life are associated with daytime sedation, those with a short half life may not affect morning vigilance. Zolpidem, which has a short half life, has not been shown to lead to an increase in daytime accidents in non insomniacs if taken at bedtime [46]. However our group of Z drugs included both Zolpidem and Zopiclone, which has an intermediate half life, and we did not control at what time treatments were taken: patients are advised to take hypnotics when they go to bed, but may also take hypnotics in the middle of the night to facilitate falling asleep following a period of wake, which reduces the time for drug metabolism. Melatonin (prescribed and OTC) was also not associated with an increase in ANMAS. Prescribed melatonin is generally slow release which is licensed in France for insomnia in patients aged over 55. Melatonin is marketed OTC in France for insomnia in formulations containing less than 2mg of normal release melatonin and is often combined with herbal medications. Patients typically take OTC or prescribed melatonin when alternative therapies such as herbal medications, homeopathy or dietary supplements have not been effective and so it is possible that they have more severe insomnia. Compared to classical hypnotics, OTC melatonin is a poor hypnotic but daytime sedation is rare which may explain its lack of effect on ANMAS [47]. We were surprised that melatonin did not offer a protective effect. Melatonin is hydroxylated by CYP1A2 in the liver and approximately 13% of the population have a slow CYP1A2 phenotype and are thus slow metabolizers of melatonin. In these subjects, melatonin levels may remain high in the morning with an

effect of vigilance [48,49]. The fact that our model analyzed the overall effect of melatonin meant that we grouped short half life OTC and slow release melatonin together. As a result, a percentage of our population taking slow release melatonin may have had a longer than expected half life with residual effects on vigilance, reducing a potential protective effect due to improved sleep. We found that alternative treatments alone (herbal medications /homeopathy/dietary supplements) were associated with a reduced risk. Alternative treatments represent a heterogenous group of treatments, most of which have not been scientifically studied. In clinical practice they are considered to have a mild effect on insomnia which may be related to the placebo effect. They are not generally associated with daytime sedation and side effects seem to be rare. The association with a reduced risk of ANMAS may be due to a modest improvement in sleep associated with a lack of daytime sedation but we note that low numbers of patients were taking only alternative treatments.

**ANMAS and sex.** The higher prevalence of driving accidents in men is well known and persists even when corrected for the higher number of male drivers and increased mileage [15,33]. However the differences in associated factors between men and women were striking and unexpected. Severe sleepiness and sleep debt increased risk in both sexes [2,3,6,7] although the odds ratios were more important in men. Gender differences were also found in insomnia with severe insomnia in men leading to an increased risk while women showed a protective effect of mild insomnia. This striking difference may be due to sleep duration. Men with insomnia are more likely to have short sleep duration than women [50]. Insomnia with short sleep duration negatively affects neurocognitive performance, especially sustained attention [51,52] and increases daytime sleepiness [50,53]. Furthermore, insomniacs with short sleep and daytime sleepiness have slowed reaction times [54], and all of the above contribute to ANMAS. Not all insomniacs have short sleep duration. Insomnia can be considered to be a disorder of hyperarousal [55] and in mild insomnia minimal sleep deprivation may be accompanied by increased vigilance during the day. In women with mild insomnia, the balance between hyperarousal and sleep deprivation may thus increase daytime vigilance leading to reduced ANMAS, whereas men, even with mild insomnia, may be more likely to be sleep deprived and are thus at increased risk. However we note that hyperarousal is not always beneficial: anxiety, which is associated with hyperarousal, leads to an increased risk of ANMAS [21] possibly due to its effects on attention. In our study significant anxiety symptoms with a high score on the HADA were only associated with ANMAS in women suggesting that their driving is disproportionately affected by the cognitive effects of anxiety. Depression is also known to increase accident risk [20], but once again we only demonstrated a link between a high HADD score and ANMAS in women. Both anxiety and depression are more common in women, but the size of our sample was sufficient to ensure adequate numbers of men and women with HADA and HADD scores >10.

Z-drugs, sedative medication and melatonin did not increase ANMAS in either group. As women on average weigh less than men but take identical drug doses, sedating effects might be expected to be more frequent in women with a potentially higher risk of ANMAS. However despite pharmacokinetic differences between men and women we did not find an increase is ANMAS in women, reflecting the findings of a study of Zolpidem which did not demonstrate differences between men and women in either efficacy or daytime sleepiness [56]. In clinical practice, women are more likely to take alternative treatments, and may automedicate with milder levels of insomnia than men. However, our model controlled for insomnia severity, and the protective effect of alternative treatments in women remains to be explained.

**Limitations.** Our study has clear limitations. Firstly, the population was recruited via a website for patients with sleep complaints. It is thus not representative of the general population, especially as hypnotic use is higher in France than in many other countries [57]. Our

results cannot be generalized to populations with a lower background level of hypnotic prescription. Secondly, accident risk over the last 6 months was based on self-report and our data does not distinguish between a near miss accident and an accident. There is a risk of recall bias for both depending on their consequences and gravity and our choice of a six month window attempted to limit this. However near miss accidents are important as they have been shown to be a risk factor for future accidents [58]. An accident or a near miss accident is not necessarily due to driver impairment and may be caused by a third party. To focus on ANMAS, we specifically asked whether the accident was attributed by the patient to their own hypovigilance. However this attribution by the participant was purely subjective and may underestimate the total number of ANMAS. Thirdly we had no way of validating hypnotic dose or the time of dose in relation to the accident or near miss accident. To minimize this, participants were asked about actual hypnotic use and ANMAS occurring over the past 6 months. Finally, it is possible that accidents or near miss accidents in patients using hypnotics may be under or over reported. Two possible mechanisms are identified: an adverse effect of hypnotics on memory and public concern about the effects of hypnotics on driving. Sedating hypnotics are known to affect memory [59,60]. However no such effects are noted for melatonin [61] which would tend to overestimate accident risk in these groups compared to Z-drugs and sedating treatments. Recent road traffic campaigns in France have focused heavily on the risk of driving and hypnotic use: it is thus possible that accidents and near miss accidents might attributed to sleepiness in patients taking hypnotics leading to an increase in ANMAS in patients taking hypnotics.

Our study provides some clear guidelines for clinicians. The risk of self reported ANMAS in patients with sleep disorders is associated with daytime sleepiness and sleep debt. Sleepy patients should be counselled concerning the potential driving risk and the importance of adequate time in bed. This risk is more marked in men in whom the presence of severe insomnia is an additional risk factor. Patient reported use of Z-drugs, sedating medication and melatonin is not associated with an increased risk of ANMAS, and in women the use of alternative OTC treatments, associated with a lower risk of ANMAS, may be preferable.

## Conclusions

Declared hypnotic use in patients suffering from sleep disorders in France is not associated with an increased risk of self reported accidents and near miss accidents attributed to sleepiness at the wheel. Men and women have different risk factor profiles: while sleepiness and sleep debt are important risk factors in both, severe insomnia increases risk in men whereas risk is reduced in women with mild insomnia and the use of alternative treatments. All patients with daytime sleepiness and men with severe insomnia should be counselled concerning driving risk and encouraged to avoid sleep debt.

## Supporting information

**S1 Data.**
(XLSX)

**S1 File. Questionnaire du sommeil.**
(DOCX)

**S2 File. Sleep questionnaire.**
(DOCX)

## Author Contributions

**Conceptualization:** Sylvie Royant-Parola, Viviane Kovess, Agnès Brion, Sylvain Dagneaux, Sarah Hartley.

**Data curation:** Sylvain Dagneaux.

**Formal analysis:** Viviane Kovess, Sarah Hartley.

**Funding acquisition:** Sylvie Royant-Parola.

**Investigation:** Sylvie Royant-Parola, Viviane Kovess, Agnès Brion, Sarah Hartley.

**Methodology:** Sylvie Royant-Parola, Agnès Brion, Sylvain Dagneaux, Sarah Hartley.

**Project administration:** Sylvain Dagneaux.

**Supervision:** Sylvie Royant-Parola.

**Validation:** Viviane Kovess, Agnès Brion, Sylvain Dagneaux, Sarah Hartley.

**Visualization:** Sarah Hartley.

**Writing – original draft:** Sylvie Royant-Parola, Viviane Kovess, Sarah Hartley.

**Writing – review & editing:** Sylvie Royant-Parola, Viviane Kovess, Agnès Brion, Sylvain Dagneaux, Sarah Hartley.

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
