## [Decision Letter · Decision Letter 0]

18 May 2020

PONE-D-19-34623

Driving accidents in patients with sleep disorders: gender affects the accident risk associated with hypnotics

PLOS ONE

Dear Dr. HARTLEY,

Thank you for submitting your manuscript to PLOS ONE. After careful consideration, we feel that it has merit but does not fully meet PLOS ONE’s publication criteria as it currently stands. Therefore, we invite you to submit a revised version of the manuscript that addresses the points raised during the review process.

In particular, You write in the title that gender affects the accident risk associated with hypnotics - and in the conclusion it is stated that hypnotics are not associated with an increased risk of ANMA. That may sound inconsistent to the title (why gender affects the accident risk associated with hypnotics if there is no risk associated with hypnotics at all?). 

Moreover, the conclusion that hypnotics do not increase the risk of accidents seems in general difficult because of the study design using self-reported questionnaires and the subjective perception. In this context it would be important to revise the title and to discuss the methodical limitations which are described in more detail by the reviewers. 

We would appreciate receiving your revised manuscript by Jul 02 2020 11:59PM. To enhance the reproducibility of your results, we recommend that if applicable you deposit your laboratory protocols in protocols.io, where a protocol can be assigned its own identifier (DOI) such that it can be cited independently in the future. For instructions see: http://journals.plos.org/plosone/s/submission-guidelines#loc-laboratory-protocols

We look forward to receiving your revised manuscript.

Kind regards,

Christian Veauthier, M.D.

Academic Editor

PLOS ONE

Journal Requirements:

Reviewers' comments:

Reviewer's Responses to Questions

**Comments to the Author**

1. Is the manuscript technically sound, and do the data support the conclusions?

Reviewer #1: No

Reviewer #2: No

Reviewer #3: Partly

2. Has the statistical analysis been performed appropriately and rigorously? 

Reviewer #1: Yes

Reviewer #2: I Don't Know

Reviewer #3: Yes

3. Have the authors made all data underlying the findings in their manuscript fully available?

Reviewer #1: Yes

Reviewer #2: Yes

Reviewer #3: Yes

4. Is the manuscript presented in an intelligible fashion and written in standard English?

Reviewer #1: Yes

Reviewer #2: No

Reviewer #3: Yes

5. Review Comments to the Author

Reviewer #1: A survey conducted on a self-selected sample of people with sleep problems. The authors investigated both road accidents and near misses. It seems that they used only one question for both phenomena, which are actually very different. Accidents have precise documentation, while near misses depend to some extent on the driver's perception. precisely for this difference, the memory of accidents can be extended to several years ago, while the memory of near misses tends to disappear quickly. Having the two phenomena mixed in the same definition of ANMA is a significant limitation of the study, which must be discussed.

A second, significant limitation, is that the authors investigated accidents and near miss "due to sleepiness", not all road accidents, thus introducing an element of accident selection entrusted to the participant, who decides whether the accident occurred or who was to happen was due to drowsiness, or not. These two unavoidable methodological flaws strongly spoil the research.

With this methodological limitation, it is inevitable to believe that the results described can be influenced by the respondents' opinions about the effects of the drugs.

In the results, the fact that the accidents attributed by drivers to sleepiness are associated with sleepiness is tautological.

Even the fact that the risk of ANMA was reduced in participants taking any medication for sleep is tautological: anyone who is taking medication to cure something is convinced that the cure works, otherwise (s)he wouldn't take the cure. So if (s)he is treating sleepiness, (s)he is convinced that sleepiness is reduced. if anyone asks him/her to indicate an accident or near miss due to the sleepiness (s)he has treated, (s)he will not report any accident. This will undoubtedly create an inverse relationship between ANMA and medication.

The discussion should be modified taking into account these important methodological limitations of the study. The claim that “accident risk (ANMA) over the preceding 6 months was increased by the presence of daytime sleepiness…” is not acceptable because the phenomenon investigated is not all accidents and near misses, but only accidents due to sleepiness. Moreover, it is not possible to know how many these accidents were and how many sensations an accident was about to happen. The evaluation of the effect of drugs is also biased by the selection of events entrusted to respondents and influenced by their opinions. Since methodological flaws cannot be remedied, the authors should express results very cautiously.

Reviewer #2: The research presents interesting results concerning gender differences in accident risk due to different sleep disorders, habits and hypnotics use. At least this is the idea which a reader gets after reading the title of the report. However, the title of the manuscript doesn’t fully correspond to the presented information and probably it should be changed after the editing of the report. It is not clear which one of the mentioned constructs is the accent – sleep disorders, gender or hypnotics? When saying driving accidents probably there should be different accidents in research, however, a single question about the presence of an accident or near-missed accident is not enough to generalize it as driving accidents.

The abstract of the report is divided into section and in general fulfil the requirements. I suggest to use another structure for the abstract: keep the given information but don’t divide it into section and make it more like a short story. The objective mentioned in the abstract is not the same at the end of the introduction – these are two different ideas, please specify.

The introduction is short in my opinion and does not include the main constructs used in the title and the objective of the study. Starting with sleepiness and then mentioning studies about hypnotics and road accidents, following by some disadvantages in the studies, then sleep disorders again. Studies on gender differences are mentioned in two lines. If this is one of the main constructs of the study, the authors should present a fuller picture concerning gender and its effect. I suggest to reorganize the introduction, start with sleepiness, continue with sleep disorders (including only these which are part of the presented study), hypnotic use, and gender differences and finally mention the disadvantages of the studies so far. You can use subtitles for all these. When putting the objective at the end of the introduction don’t forget to specify some of the hypothesis which you have because they are missing. The main aim along with the hypothesis should be in subtitle again.

The second section of the manuscript should be named Materials and Methods and it usually starts with the description of the Sample (first subtitle), so I suggest to move this paragraph from Results here (the sample is not results). Also, when describing the sample you should mention the gender distribution, age (range, mean) and also the main characteristics of the sample which are important for the study. It is not necessary to put the information in a table and also avoid info which is not important (the profession is not in the objective or the results, so why are you mention it?). The table in the Sample paragraph is too big and different for understanding. The first paragraph here is probably the Procedure (third subtitle) of the study and the second paragraph (now Variables) should be Methods (second subtitle) which have to be fully presented including information of the whole consistency of the instruments. When saying that this is observational study what exactly do you mean? As I understand this is an online questionnaire with self-reported answers. The last subtitle is the last paragraph here, concerning the data analysis and how they were performed.

The Results section should include only results concerning the presented hypothesis and the main objective. The distribution of the sample according to the investigated constructs should be presented in the Sample paragraph, not in the Results. The whole Result section is really messy. It is not clear what the accent is again! When specifying the main it would be easier to construct a good Result section. Table 2 and 3 are unnecessary in this current manuscript. I guess that you are trying to study the connections between sleep disorders, hypnotic use, gender and ANMA. If this is the case the Result section should include at least 3 subtitles presenting the association between sleep disorders and ANMA, hypnotic use and ANMA, gender and ANMA. If Depression and Anxiety are studied in association with the ANMA and you really want to present the results in this manuscript, you should put these constructs in the objective, or in the hypothesis, but you can always use these analyses in another article. Also if you are presenting info about habits, you should also mention that in the Methods paragraph (but in my opinion this may be used in another article again). Any other information here is additional and may only make the manuscript hard for understanding.

The Discussion section should follow the results. Any specifics about methods or sample shouldn’t be here. Here you say you have unexpected findings, but what were the expected ones, is not clear (hypothesis). Put the Limitations of the study in the separate subtitle.

Reviewer #3: The authors aimed to evaluate driving accidents in patients with sleep disorders and assessing accident risk associated with hypnotics.

The main conclusion is that the “hypnotic use in patients suffering from sleep disorders in France is NOT associated with an increased risk of accidents and near miss accidents attributed to sleepiness at the wheel”.

While we agree that patients suffering from insomnia are hypervigilant, given the lack of information about medication dose or the timing the current conclusion seems like an over statement, and may even send a wrong message to the general population. Additionally, given the nature of study (self-reported), concluding that hypnotics do not increase the risk of accidents is not accurate. The authors have mentioned this in the limitation, but I recommend re stating the conclusion.

6. PLOS authors have the option to publish the peer review history of their article (what does this mean?). If published, this will include your full peer review and any attached files.

Reviewer #1: Yes: Nicola Magnavita

Reviewer #2: No

Reviewer #3: No

---

## [Author Response · Author response to Decision Letter 0]

12 Jun 2020

Dear Dr Veauthier,

Thank you so much for your reply in this difficult period for health care workers. We would particularly like to thank your reviewers for their careful reading of our article and their helpful comments. 

In response to your general comments:

We agree that our title was not clear and we have modified it to better reflect the content of the article. 

Do hypnotics increase the risk of driving accidents or near miss accidents due to hypovigilance? The effects of sex, chronic sleepiness, sleep habits and sleep pathology. 

Data sharing: we will upload the database used for the study

We have replied point by point to the comments of your reviewers: in the attached word document, our comments appear in red and modifications to the text are indicated in italics.

 

Reviewer #1: A survey conducted on a self-selected sample of people with sleep problems. The authors investigated both road accidents and near misses. It seems that they used only one question for both phenomena, which are actually very different. Accidents have precise documentation, while near misses depend to some extent on the driver's perception. precisely for this difference, the memory of accidents can be extended to several years ago, while the memory of near misses tends to disappear quickly. Having the two phenomena mixed in the same definition of ANMA is a significant limitation of the study, which must be discussed. 

We agree that the accidents and near miss accidents are different phenomena. The term “accident” covers everything from a scratched bumper to a major accident with injury, and “near miss accidents” can equally be minor or major ; both of them can be forgotten depending on their consequences and gravity. Our question asked specifically whether participants had an accident or near miss accident over the preceding 6 months that was attributable to sleepiness at the wheel, in an attempt to limit the problems of recall. Our study did not aim to examine the prevalence of ANMA but to examine factors associated with ANMA. We have modified the text in the limitations section as follows:

Secondly, accident risk over the last 6 months was based on self-report and our data does not distinguish between a near miss accident and an accident. There is a risk of recall bias for both depending on their consequences and gravity and our choice of a six month window attempted to limit this. However near miss accidents are important as they have been shown to be a risk factor for future accidents[52].

A second, significant limitation, is that the authors investigated accidents and near miss "due to sleepiness", not all road accidents, thus introducing an element of accident selection entrusted to the participant, who decides whether the accident occurred or who was to happen was due to drowsiness, or not. These two unavoidable methodological flaws strongly spoil the research.

We agree with our reviewer that we are only studying ANMA due to sleepiness. We did not record ANMA from other causes and our findings do not apply to all ANMA. We have changed the title of our article to reflect this.

Do hypnotics increase the risk of driving accidents or near miss accidents due to hypovigilance? The effects of sex, chronic sleepiness, sleep habits and sleep pathology.

 To further reinforce the message we have used the term ANMAS (ANMA “due to sleepiness”) throughout the text and modified the definition of ANMA given in the methods section as follows:

Accident risk was explored by asking whether the patient had had a driving Accident or a Near Miss Accident due to Sleepiness (ANMAS) over the past six months.

ANMA due to sleepiness remain an important public health topic: with the reduction of speeding and driving under the influence of alcohol thanks to road safety campaigns, accidents have been reduced in France, but of those that remain it is estimated that up tp 37% are due to reduced vigilance at the wheel. Our study attempted to examine these accidents, notably because a link has been suggested between the use of hypnotics and reduced vigilance. We have modified the introduction to stress the importance of accidents related to hypovigilance:

Sleepiness at the wheel is an important cause of traffic accidents, with estimates of the proportion of accidents due to hypovigilance ranging from 3 – 37%. The relative proportion of accidents due to hypovigilance in France is increasing as road safety campaigns sucessfully reduce speeding and driving under the influence of alcohol and drugs[1–3].

We also agree with the limitations of using subjective reporting of sleepiness as a cause of accident or near accident. This is a personal attribution which depends on the participants perception of the cause of the ANMA. We note that other questionnaire studies have also relied on self report. Our question is the same as that used in two preceding studies in France (Quera-Salva 2006, Philip 1999), but we reduced the time frame to 6 months rather than a year specifically to address issues of recall. We have modified the limitations section to reflect this:

To focus on ANMAS, we specifically asked whether the accident was attributed by the patient to their own hypovigilance. However this attribution by the participant was purely subjective and may underestimate the total number of ANMAS.

With this methodological limitation, it is inevitable to believe that the results described can be influenced by the respondents' opinions about the effects of the drugs. In the results, the fact that the accidents attributed by drivers to sleepiness are associated with sleepiness is tautological. 

This is an important point. We agree with the reviewer that there is a link between the hypovigilance at the moment of an ANMAS which refers to a specific point in time and the experience of chronic sleepiness: indeed this is exactly what our results show. 

We chose to measure chronic sleepiness using the Epworth Sleepiness Scale (ESS) which is a well known, validated scale (Johns 1991) and which has been extensively used in previous studies of accidents (Goncalves 2015, Stutts 2003, Philip 1996, Quera-Salva 2006). These studies also show that while the risk of accident is increased in chronically sleepy patients, not all patients reporting ANMAS suffer from chronic sleepiness, but find that other factors such as acute sleep debt (Philip 1996) may be relevant. Our finding that ANMAS are linked strongly to chronic sleepiness thus confirms the findings of previous studies. We have added the information about the link with chronic sleepiness in the discussion.

We found that ANMAS were strongly linked to chronic sleepiness as measured by the ESS,

Even the fact that the risk of ANMA was reduced in participants taking any medication for sleep is tautological: anyone who is taking medication to cure something is convinced that the cure works, otherwise (s)he wouldn't take the cure. So if (s)he is treating sleepiness, (s)he is convinced that sleepiness is reduced. if anyone asks him/her to indicate an accident or near miss due to the sleepiness (s)he has treated, (s)he will not report any accident. This will undoubtedly create an inverse relationship between ANMA and medication.

This is an key question: the underlying argument is that if people are aware that they are taking drugs which cure hypovigilance, they assume that they cannot have ANMAS which are due to hypovigilance. If this was the case, patients taking hypnotics would not report ANMAS, but our data show that patients on hypnotics do indeed report ANMAS. Current road safety campaigns in France have focused heavily on the risks of medication and driving, and patients are aware that hypnotics are considered to increase the risk of accident ; this would potentially promote the reverse bias. As a consequence the direction of the effect is unknown. We have acknowledged this in the limitations section. 

Finally it is possible that accidents or near miss accidents in patients using hypnotics may be under or over reported. Two possible mecanisms are identified: an adverse effect of hypnotics on memory and public concern about the effects of hypnotics on driving. …… Recent road traffic campaigns in France have focused heavily on the risk of driving and hypnotic use: it is thus possible that accidents and near miss accidents might attributed to sleepiness in patients taking hypnotics leading to an increase in ANMAS in patients taking hypnotics.

The discussion should be modified taking into account these important methodological limitations of the study. The claim that “accident risk (ANMA) over the preceding 6 months was increased by the presence of daytime sleepiness…” is not acceptable because the phenomenon investigated is not all accidents and near misses, but only accidents due to sleepiness. 

We agree with the reviewer and have modified our discussion extensively to make it clear that we are focussing purely on ANMA due to hypovigilance (ANMAS)

Moreover, it is not possible to know how many these accidents were and how many sensations an accident was about to happen. The evaluation of the effect of drugs is also biased by the selection of events entrusted to respondents and influenced by their opinions. Since methodological flaws cannot be remedied, the authors should express results very cautiously. 

We agree completely with the reviewer: our study was not designed to study the prevalence of ANMA, only the factors linked with ANMAS. We appreciate the reviewer’s advice concerning caution in our conclusions. We have discussed the subjective nature of our findings in the limitations section, specifically in relation to the reporting of hypnotic use

Thirdly we had no way of validating hypnotic dose or the time of dose in relation to the accident or near miss accident. To minimize this, participants were asked about actual hypnotic use and ANMAS occurring over the past 6 months. Finally it is possible that accidents or near miss accidents in patients using hypnotics may be under or over reported. Two possible mecanisms are identified: an adverse effect of hypnotics on memory and public concern about the effects of hypnotics on driving. Sedating hypnotics are known to affect memory [59,60]. However no such effects are noted for melatonin[61] which would tend to overestimate accident risk in these groups compared to Z-drugs and sedating treatments. Recent road traffic campaigns in France have focused heavily on the risk of driving and hypnotic use: it is thus possible that accidents and near miss accidents might attributed to sleepiness in patients taking hypnotics leading to an increase in ANMAS in patients taking hypnotics. 

We have modified the final paragraph of our discussion as follows:

Our study provides some clear guidelines for clinicians. The risk of ANMAS in patients with sleep disorders is associated with daytime sleepiness and sleep debt, and sleepy patients should be counselled concerning the potential driving risk and the importance of adequate time in bed. This risk is more marked in men in whom the presence of severe insomnia is an additional risk factor. The use of Z-drugs, sedating medication and melatonin is not associated with an increased risk of ANMAS, and in women the use of alternative OTC treatments, associated with a lower risk of ANMAS, may be preferable.

 

Reviewer #2: The research presents interesting results concerning gender differences in accident risk due to different sleep disorders, habits and hypnotics use. At least this is the idea which a reader gets after reading the title of the report. However, the title of the manuscript doesn’t fully correspond to the presented information and probably it should be changed after the editing of the report. It is not clear which one of the mentioned constructs is the accent – sleep disorders, gender or hypnotics? 

We agree with the reviewer concerning the title and have changed it to better reflect the content of the article

Do hypnotics increase the risk of driving accidents or near miss accidents due to hypovigilance? The effects of sex, chronic sleepiness, sleep habits and sleep pathology. 

When saying driving accidents probably there should be different accidents in research, however, a single question about the presence of an accident or near-missed accident is not enough to generalize it as driving accidents.

We agree and have modified the title. We have also changed the acronym ANMA to underline the fact that the term relates to accidents and near miss accidents attributed to sleepiness (ANMAS)

The abstract of the report is divided into section and in general fulfil the requirements. I suggest to use another structure for the abstract: keep the given information but don’t divide it into section and make it more like a short story. 

We have rewritten the abstract in a narrative form without subheadings.

Driving accidents due to hypovigilance are common but the role of hypnotics is unclear in patients suffering from sleep disorders. Our study examined factors influencing accidents and near miss accidents attributed to sleepiness at the wheel (ANMAS). Using data from an online questionnaire aimed at patients with sleep disorders, we analysed the associations between ANMAS, sociodemographic data, symptoms of sleep disorders, severity of insomnia (Insomnia Severity Index (ISI)) symptoms of anxiety and depression (Hospital Anxiety and Depression scale with depression (HADD) and anxiety (HADA) subscales), chronic sleepiness (Epworth sleepiness scale ESS), hypnotic use and information about sleep habits. Hypnotics were hierarchically grouped into Z-drugs, sedative medication, melatonin and over the counter (OTC) alternative treatments.

Of 10802 participants; 9.1% reported ANMAS (Men 11.1% women 8.3%) and 24.4% took hypnotics (Z-drugs 8.5%, sedative medication 8%, melatonin 5.6% and alternative treatments 2.5%). Logistic regression analysis identified the following risk factors for ANMAS: moderate (OR 2.4; CI: 2.10-2.79) and severe sleepiness (ESS OR 5.66; CI: 4.74-6.77), depression (HADD OR 1.2; CI: 1.03-1.47), anxiety (HADA OR 1.2;CI: 1.01-1.47), and insufficient sleep (OR1.4; CI: 1.2-1.7). Hypnotics were not associated with an increased risk of ANMAS in patients suffering from insomnia. Risk factors varied according to sex: in females, sex (OR 0.; CI: 0.55-0.74), mild insomnia (OR 0.5; CI: 0.3-0.8) and use of alternative treatments (OR 0.455, CI:0.23-0.89) were protective factors and risk was increased by sleepiness, sleep debt, social jetlag, caffeine use, anxiety and depression. In men no protective factors were identified: sleepiness, sleep debt, and severe insomnia were associated with an increased risk of ANMAS. In clinical practice, all patients with daytime sleepiness and men with severe insomnia should be counselled concerning driving risk and encouraged to avoid sleep debt.

The objective mentioned in the abstract is not the same at the end of the introduction – these are two different ideas, please specify.

We have clarified the objective in the introduction as follows 

Our objective was to explore factors associated with accidents and near miss accidents attributed to sleepiness at the wheel (ANMAS)

The introduction is short in my opinion and does not include the main constructs used in the title and the objective of the study. Starting with sleepiness and then mentioning studies about hypnotics and road accidents, following by some disadvantages in the studies, then sleep disorders again. Studies on gender differences are mentioned in two lines. If this is one of the main constructs of the study, the authors should present a fuller picture concerning gender and its effect. I suggest to reorganize the introduction, start with sleepiness, continue with sleep disorders (including only these which are part of the presented study), hypnotic use, and gender differences and finally mention the disadvantages of the studies so far. You can use subtitles for all these. 

We have extensively rewritten the introduction, using subheadings in the proposed format

When putting the objective at the end of the introduction don’t forget to specify some of the hypothesis which you have because they are missing. The main aim along with the hypothesis should be in subtitle again.

We have rewritten the hypothesis and objectives and introduced a subtitle

Hypothesis and objective:

Hypothesis and aim:

Our hypothesis was that factors known to be associated with increased daytime sleepiness such as sleep apnea, sleep debt and the use of hypnotics would be associated with ANMAS.

Our objective was to explore factors associated with accidents and near miss accidents attributed to sleepiness at the wheel (ANMAS). 

We have reorganised the information in the introduction as suggested by the reviewer and introduced subtitles to aid the reader. 

The second section of the manuscript should be named Materials and Methods and it usually starts with the description of the Sample (first subtitle), so I suggest to move this paragraph from Results here (the sample is not results). Also, when describing the sample you should mention the gender distribution, age (range, mean) and also the main characteristics of the sample which are important for the study. It is not necessary to put the information in a table and also avoid info which is not important (the profession is not in the objective or the results, so why are you mention it?). The table in the Sample paragraph is too big and different for understanding. 

We have rewritten the materials and methods section following the suggestions of the reviewer and reduced the information in table 1 by removing the information about professions.

The first paragraph here is probably the Procedure (third subtitle) of the study and the second paragraph (now Variables) should be Methods (second subtitle) which have to be fully presented including information of the whole consistency of the instruments. When saying that this is observational study what exactly do you mean? As I understand this is an online questionnaire with self-reported answers. The last subtitle is the last paragraph here, concerning the data analysis and how they were performed.

We have reorganised the information about the procedure and the methods, including references about the development of the validated scales used (ESS, ISI HAD) and added information about thresholds.

The Results section should include only results concerning the presented hypothesis and the main objective. The distribution of the sample according to the investigated constructs should be presented in the Sample paragraph, not in the Results. The whole Result section is really messy. It is not clear what the accent is again! When specifying the main it would be easier to construct a good Result section. Table 2 and 3 are unnecessary in this current manuscript. I guess that you are trying to study the connections between sleep disorders, hypnotic use, gender and ANMA. If this is the case the Result section should include at least 3 subtitles presenting the association between sleep disorders and ANMA, hypnotic use and ANMA, gender and ANMA.

We have reorganised and rewritten the results section. We have removed the data presented in tables 2 and 3 and also references to this data in the discussion section

If Depression and Anxiety are studied in association with the ANMA and you really want to present the results in this manuscript, you should put these constructs in the objective, or in the hypothesis, but you can always use these analyses in another article. 

Anxiety and depression are highly prevalent in sleep disorders and are also shown to be independent risk factors for ANMAS. In order to differentiate between the different elements, an analysis including anxiety and depression was essential and we have retained the data. We have added information in the introduction to explain the importance of psychiatric disorders. 

Both anxiety and depression are prevalent in patients with sleep disorders but have also been shown to independently increase accident risk[20,21].

Also if you are presenting info about habits, you should also mention that in the Methods paragraph (but in my opinion this may be used in another article again). Any other information here is additional and may only make the manuscript hard for understanding.

We feel that retaining information about sleep habits is essential: they have been repeatedly identified as important in epidemiological studies, with a clear increase in accident risk in patients who have an acute or chronic sleep debt.

The Discussion section should follow the results. Any specifics about methods or sample shouldn’t be here. Here you say you have unexpected findings, but what were the expected ones, is not clear (hypothesis). Put the Limitations of the study in the separate subtitle.

We have redrafted and reorganised the discussion section, with our initial hypothesis clearly stated at the beginning of the discussion, eliminated data on the links between hypnotics and sleepiness and provided clear subheadings for ease of reading.

 

Reviewer #3: The authors aimed to evaluate driving accidents in patients with sleep disorders and assessing accident risk associated with hypnotics.

The main conclusion is that the “hypnotic use in patients suffering from sleep disorders in France is NOT associated with an increased risk of accidents and near miss accidents attributed to sleepiness at the wheel”.

While we agree that patients suffering from insomnia are hypervigilant, given the lack of information about medication dose or the timing the current conclusion seems like an over statement, and may even send a wrong message to the general population. 

We agree that the conclusions of this study need to be carefully worded and that the lack of information about the precise link between the timing of hypnotic use and ANMAS has not been explored. 

Additionally, given the nature of study (self-reported), concluding that hypnotics do not increase the risk of accidents is not accurate. The authors have mentioned this in the limitation, but I recommend re stating the conclusion.

We have redrafted the conclusion to reflect the element of self report.

Declared hypnotic use in patients suffering from sleep disorders in France is not associated with an increased risk of self reported accidents and near miss accidents attributed to sleepiness at the wheel.

Thank you once again for your time spent in reading and improving our article. 

Yours sincerely

Sarah Hartley

Viviane Kovess

Sylvie Royant Parola

---

## [Decision Letter · Decision Letter 1]

8 Jul 2020

Do hypnotics increase the risk of driving accidents or near miss accidents due to hypovigilance? The effects of sex, chronic sleepiness, sleep habits and sleep pathology.

PONE-D-19-34623R1

Dear Dr. HARTLEY,

We’re pleased to inform you that your manuscript has been judged scientifically suitable for publication and will be formally accepted for publication once it meets all outstanding technical requirements.

Kind regards,

Christian Veauthier, M.D.

Academic Editor

PLOS ONE

Reviewers' comments:

Reviewer's Responses to Questions

**Comments to the Author**

1. If the authors have adequately addressed your comments raised in a previous round of review and you feel that this manuscript is now acceptable for publication, you may indicate that here to bypass the “Comments to the Author” section, enter your conflict of interest statement in the “Confidential to Editor” section, and submit your "Accept" recommendation.

Reviewer #1: All comments have been addressed

Reviewer #2: All comments have been addressed

2. Is the manuscript technically sound, and do the data support the conclusions?

Reviewer #1: Yes

Reviewer #2: Yes

3. Has the statistical analysis been performed appropriately and rigorously? 

Reviewer #1: Yes

Reviewer #2: Yes

4. Have the authors made all data underlying the findings in their manuscript fully available?

Reviewer #1: (No Response)

Reviewer #2: Yes

5. Is the manuscript presented in an intelligible fashion and written in standard English?

Reviewer #1: Yes

Reviewer #2: Yes

6. Review Comments to the Author

Reviewer #1: the manuscript has been improved according to authors' possibilities, the data obtained could be useful

Reviewer #2: All of the comments were take into account by the authors, the manuscript is fully edited and now is much easier for reading and understanding. The text is well written and the presented data supports the conclusions.

7. PLOS authors have the option to publish the peer review history of their article (what does this mean?). If published, this will include your full peer review and any attached files.

Reviewer #1: **Yes: **Nicola Magnavita

Reviewer #2: No

---

## [Editor Report · Acceptance letter]

16 Jul 2020

PONE-D-19-34623R1 

Do hypnotics increase the risk of driving accidents or near miss accidents due to hypovigilance? The effects of sex, chronic sleepiness, sleep habits and sleep pathology. 

Dear Dr. Hartley:

I'm pleased to inform you that your manuscript has been deemed suitable for publication in PLOS ONE. Congratulations! Your manuscript is now with our production department. 

Kind regards, 

on behalf of

Dr. Christian Veauthier 

Academic Editor

PLOS ONE